# Discovery of Phenylcarbamoylazinane-1,2,4-Triazole Amides Derivatives as the Potential Inhibitors of Aldo-Keto Reductases (AKR1B1 & AKRB10): Potential Lead Molecules for Treatment of Colon Cancer

**DOI:** 10.3390/molecules27133981

**Published:** 2022-06-21

**Authors:** Amna Saeed, Syeda Abida Ejaz, Muhammad Sarfraz, Nissren Tamam, Farhan Siddique, Naheed Riaz, Faizan Abul Qais, Samir Chtita, Jamshed Iqbal

**Affiliations:** 1Department of Pharmaceutical Chemistry, Faculty of Pharmacy, The Islamia University of Bahawalpur, Bahawalpur 63100, Pakistan; amna.saeed757@gmail.com; 2College of Pharmacy, Al Ain Campus, Al Ain University, Al Ain P.O. Box 64141, United Arab Emirates; muhammad.sarfraz@aau.ac.ae; 3Department of Physics, College of Science, Princess Nourah bint Abdulrahman University, P.O Box 84428, Riyadh 11671, Saudi Arabia; nmtamam@pnu.edu.sa; 4Laboratory of Organic Electronics, Department of Science and Technology, Linköping University, SE-60174 Norrköping, Sweden; drfarhansiddique@gmail.com; 5Department of Pharmacy, Royal Institute of Medical Sciences (RIMS), Multan 60000, Pakistan; 6Department of Chemistry, Baghdad-ul-Jadeed Campus, The Islamia University of Bahawalpur, Bahawalpur 63100, Pakistan; nrch322@yahoo.com; 7Department of Agricultural Microbiology, Faculty of Agricultural Sciences, Aligarh Muslim University, Aligarh 202002, UP, India; faizanabulqais@gmail.com; 8Laboratory of Analytical and Molecular Chemistry, Faculty of Sciences Ben M’Sik, Hassan II University of Casablanca, Sidi Othmane, Casablanca BP7955, Morocco; samirchtita@gmail.com; 9Centre for Advanced Drug Research, Abbottabad Campus, COMSATS University Islamabad, Abbotabad 22060, Pakistan

**Keywords:** aldose reductase, molecular docking, density functional theory, ADMET properties, AutoDock tools, Molecular Operating Environment

## Abstract

Both members of the aldo-keto reductases (AKRs) family, AKR1B1 and AKR1B10, are over-expressed in various type of cancer, making them potential targets for inflammation-mediated cancers such as colon, lung, breast, and prostate cancers. This is the first comprehensive study which focused on the identification of phenylcarbamoylazinane-1, 2,4-triazole amides (**7a**–**o**) as the inhibitors of aldo-keto reductases (AKR1B1, AKR1B10) via detailed computational analysis. Firstly, the stability and reactivity of compounds were determined by using the Guassian09 programme in which the density functional theory (DFT) calculations were performed by using the B3LYP/SVP level. Among all the derivatives, the **7d**, **7e**, **7f**, **7h**, **7j**, **7k**, and **7m** were found chemically reactive. Then the binding interactions of the optimized compounds within the active pocket of the selected targets were carried out by using molecular docking software: AutoDock tools and Molecular operation environment (MOE) software, and during analysis, the Autodock (academic software) results were found to be reproducible, suggesting this software is best over the MOE (commercial software). The results were found in correlation with the DFT results, suggesting **7d** as the best inhibitor of AKR1B1 with the energy value of −49.40 kJ/mol and **7f** as the best inhibitor of AKR1B10 with the energy value of −52.84 kJ/mol. The other potent compounds also showed comparable binding energies. The best inhibitors of both targets were validated by the molecular dynamics simulation studies where the root mean square value of <2 along with the other physicochemical properties, hydrogen bond interactions, and binding energies were observed. Furthermore, the anticancer potential of the potent compounds was confirmed by cell viability (MTT) assay. The studied compounds fall into the category of drug-like properties and also supported by physicochemical and pharmacological ADMET properties. It can be suggested that the further synthesis of derivatives of **7d** and **7f** may lead to the potential drug-like molecules for the treatment of colon cancer associated with the aberrant expression of either AKR1B1 or AKR1B10 and other associated malignancies.

## 1. Introduction

Colon cancer is the third most prevalent type of cancer in the world [1], and is the third most common cancer in the United States. Several factors, including familial and hereditary factors, play a key role in colon carcinogenesis, which has a higher prevalence in males than in women [2]. Cells that have sustained irreparable DNA damage as a result of increased free radical production, a reduction in extracellular stimulation, which regulates cell growth, and the transmission of cancer genes through autosomal dominant inheritance are among the most important initiators of carcinogenesis. Despite the fact that the relationship between inflammation and colon cancer is not completely understood, data and already reported evidence strongly support its correlation [3].

The enzyme aldo-keto reductase (AKR), which is closely related to aldose reductase (AR), is widely found in humans and is categorized into structural and functional families: aldehyde reductase (AKR1A), the AR (AKR1B), the aldo-keto reductase family 1 member B1 (AKR1B1), the aldo-keto reductase family 1 member B10 (AKR1B10), the aflatoxin reductase (AKR7A), and the keto-steroid (oxosteroid) reductase (AKR1D) [4]. These aldo-keto reductases are involved in various biochemical pathways such as steroid conversion, fructose synthesis, osmoregulation, aldehyde detoxification, and catecholamine metabolism, which are required for normal physiological functions [5].

Human aldo-keto reductase family 1 member B (AKR1B) is one of several subtypes of aldo-keto reductase that can be found in several organ tissues, including the liver, lungs, breast, pancreas, and small intestine [6]. AKR1B1 is also known as human aldose reductase, whereas AKR1B10 is known as human small intestine reductase [7]. The former is produced in the prostate, skeletal muscle, adrenal gland, and heart [8].

It has been established that aldose reductase (AKR1B1) functions as an essential mediator of oxidative stress and inflammation caused by growth hormones, cytokines, and carcinogens [1]. Cancer development and metastasis are both aided by the stimulation of nuclear factor-kB signaling (NF-kB) by reactive oxygen species (ROS) [9]. AR (AKR1B1) suppression has the potential to be a unique treatment strategy for the prevention of colon cancer metastasis. It also modulates cancer cell adhesion, invasion, and migratory processes, all of which are associated with metastasis. This requires the establishment of adequate blood flow to the tumor cells in order for the malignancy to metastasize to other organs [10,11].

Another member of the family AKR, i.e., AKR1B10, is known as the antineoplastic target. It has been used as a cancer biomarker, as it reduces retinal to retinol, which leads to lower levels of retinoic acid because retinal is the source of retinoic acid, which ultimately effects the proliferation and differentiation involved in carcinogenesis [8]. AKR1B10 shows great similarity (71% amino acid sequence) in structure with AKR1B1. That is why for the identification of selective inhibitors, both enzymes are emergent targets [12,13]. Although there is a wealth of literature available on anticancer drugs [14,15,16,17], no drug has yet been identified for the selected targeted proteins. Therefore, searching for selective inhibitors for these two targets is a great challenge. Figure 1 below illustrates the correlation between inflammation mechanism and aldo-keto reductases (AKR1B1, AKR1B10).

Many research groups have predicted the inhibitory effect of various classes of heterocyclic compounds on the other isoforms of aldose reductase that are involved in diabetes [19,20] and cardiac complications [21,22], i.e., ALR-1 and ALR-2, but none of them have been discussed against the selected targeted proteins, i.e., AKR1B1 and AKR1B10, which are specifically involved in colon cancer.

Numerous studies have endorsed the promising fact that inhibition of aldose reductase plays a vital role in treating colon cancer [1,8,23], as well as diabetic and cardiovascular complications [24,25].

Figure 2 illustrates the selected examples of aldose reductase inhibitors: (a) Acetic acid derivatives (epalrestat), (b) spiro hydantoins (cyclic imides; sorbinil), and (c) fidarestat.

Moreover, the previous enzyme inhibition study by our group [27] was performed only using 15-lipoxygenase enzyme, from soybean source, indicating compounds have anti-inflammatory potential. The current findings aim to discover the new and small molecules as potent human aldose reductase inhibitors that may help to synthesize more effective molecules with drug-like properties for the treatment of colon cancer that specifically occurs from the aberrant expression of either protein (Figure 1) To extend our findings, in current research study, we considered the previously synthesized phenylcarbamoylazinane-1,2,4-triazole amides derivatives (**7a**–**o**) with anti-inflammatory activity by our group [27], against two enzymes of aldo-keto reductase, i.e., AKR1B1 and AKR1B10 via in silico studies.

The binding interactions of all these derivatives were determined by using two molecular docking programs; one was academic, i.e., AutoDock (version 1.5.6), and one was commercial, i.e., Molecular Operating Environment (MOE Dock version 2015.10). The purpose of using two software was to validate the results of binding interactions. The results were further validated by detailed quantum chemistry calculations, i.e., density functional theory calculations (DFTs) in which frontier molecular orbitals (FMOs), global and local reactivity descriptors, and molecular electrostatic potential (MEP) were calculated. The detailed drug-like properties, i.e., ADMET properties of the compounds, were performed to support the above studies. This is the first comprehensive computational-based study that may lead to the findings and discovery of potent and selective inhibitors of either human aldose reductase (AKR1B1) or human small intestine reductase (AKR1B10), suggesting these derivatives for further exploration at the molecular level as ideal candidates for colon cancer and other associated malignancies.

## 2. Experimental

### 2.1. Instrumentation and Methods

The chemicals and solvents were purchased from Sigma, Aldrich, and Alfa Aesar. To get ^1^H and ^13^C NMR spectra, tetramethylsilane was used as an internal reference, and the experiment was performed on a Bruker instrument. KBr pellets were IR spectra on a Shimadzu 460 FTIR spectrometer. Mass spectra were achieved using a data system and a JMSA 500 mass spectrometer. Melting points were determined by using Gallen Kemp electrothermal apparatus.

### 2.2. Chemistry

#### Synthesis of Phenylcarbamoylazinane-1,2,4-Triazole Amides Derivatives

A general procedure of the synthesis has been reported in our previous work [27].

### 2.3. Density Functional Theory (DFT)

The most efficient and popular method, density functional theory (DFT), in quantum chemistry was implemented to have a better insight of molecular properties of the compounds. It was performed for computing number of information following; geometry optimization, FMO, global and local reactivity descriptors, MEP. All the calculations were carried out using the Guassian09 program [28] where Becke-3-Parameter-Lee-Yang-Parr (B3LYP) in SVP basis set [29] was used. At the end, the output check files were visualized and interpreted via Gauss View version 5 [30,31,32].

### 2.4. Molecular Docking

#### 2.4.1. Preparation of Ligands

Ligands were prepared by using AutoDock (version 1.5.6); one was commercial, i.e., Molecular Operating Environment (MOE Dock version 2015.10). The structures of a total of 15 compounds were drawn in ChemDraw 12 ultra [33]. The compounds (**7a**–**7o**) were converted to 3D structure (.pdb) and (.sdf) files by using Chem3D pro 12 [34]. These compounds were saved into pdb and sdf formats by Chem3D pro 12 after energy minimization.

#### 2.4.2. Preparation of Target

The targeted proteins, AKR1B1 and AKR1B10, which were redeemed from the RCSB Protein Data Bank [35], AKR1B1 (PDB ID: 6f7r) and AKR1B10 (PDB ID: 4gqg), helped as docking receptors. In preparing the protein, all the hetatoms and water molecules were removed from the protein both in the Autodock [36] and Molecular Operation Environment (MOE) [37]. Polar hydrogen atoms and Kollman charges were added while using both software.

#### 2.4.3. Molecular Docking Protocol

The compounds (**7a**–**7o**) were docked using AutoDock using the standard procedure within the active site of proteins AKR1B1 and AKR1B10. In AutoDock, the grid box was built using a size 60 × 60 × 60 Å, pointing in x, y, and z coordinates, respectively, with a grid point spacing of 0.375 Å. The centre of the grid box was adjusted to cover the active pocket. In MOE, dummies were created at the active site.

For AutoDock, the no. of population was set to 150 and the no. of genetic algorithm (GA) runs were set to 100. For pose scoring, default scoring function was used. For MOE, MMFF94x forcefield was loaded [38]. The default placement method, Triangle Matcher algorithm, was implemented for generating pose. Two rescoring functions included London dG and GBVI/WSA dG, and they were implemented for pose scoring [39,40].

#### 2.4.4. Visualization

In AutoDock, visualization was carried out using Discovery studio visualizer (version 2020) [41]. Two-dimensional and three-dimensional interactions between protein and ligand were analyzed. In MOE, 2D and 3D interactions were interpreted in MOE window by opening the output file.

The conformations with the top score (minimum) binding energy were selected for examining the interactions between the target (protein) and ligands [42].

#### 2.4.5. Validation

The final assessment was done by calculating the RMSD value. From 100 runs, one docking pose was considered successful, whose RMSD value was between the docking pose and the experimentally determined conformation of a ligand was ≤2.0 Å [43]. As a consequence, RMSD is the most accurate and best way of computing the best pose, but an acceptable one, as no other practical validation methods are available [44].

**Vincristine**, a known anti-cancer drug, was docked to validate our studies. Another way used to validate our studies was the docking of co-crystal ligand **NAP** within the active binding pocket (to reproduce the same configuration).

### 2.5. Molecular Dynamics Simulations

Gromacs-2018.1 with the am-ber99sb-ILDN force field was used for the simulation studies of the AKR1B1, AKR1B10 protein and their complexes (in an aqueous environment) [45,46]. The AM1-BCC charge model and the Antechamber packages in AmberTools21 were used to build the topology for both ligands after they were extracted from their respective complexes [47]. The TIP3P water model was used to solvate both proteins alone and in complexes, and then counter sodium/chlorine ions were introduced to neutralize their charges. Highly restrictive descent minimization was used to limit all systems to a maximum of 50,000 steps in order to exclude any weak van der Waals connections. Using a V-rescale thermostat set at 300 K and a constant volume throughout one nanosecond, all systems were initially equilibrated (NVT equilibration) [48]. A Parrnel-lo-Rahman barostat was used to complete the second equilibration at 1.0 bar and 300 K for one nanosecond (NPT equilibration). Both proteins and their complexes were simulated at 100 ns, and ten thousand frames of each trajectory were recorded [49]. PBC changes were made to the trajectory prior to the analysis. The MM-PBSA was used to calculate the ligand-protein binding energies [50].

### 2.6. Cell Viability Assay

To assess the anti-cancer capability of the compounds, the cell viability assay of the most potent derivative was estimated against human cervical cancer cell line (HeLa). The experiment was performed according to the reported method Mosmann (in 1983) and Nikš and Otto (in 1990), respectively [51,52]. The experiment was performed in 96–well flat–bottom plates in 90 μL of medium containing 10 × 10^4^ cells, seeded into each well. The 100 µL of test compound solution was added to the respective well and the plate was allowed for 24 h of incubation at 37 °C and 5% CO_2_. The positive and negative control wells were seeded with 10 μL of standard drug (cisplatin) and 100 μL of cell media (no compound), respectively. Each well was then pipetted with 10 μL of MTT reagent and incubated for 4 h at 37 °C. Then, 100 µL of 10% sodium dodecyl sulphate solution was added and kept at room temperature for 30 min. with occasional shaking. Finally, optical density was calculated. The ability of mitochondrial dehydrogenase to generate formazan complex indicated the presence of metabolically active cells (viable cells). All studies were conducted in triplicate, and the results were reported as percent growth inhibition values, as reported earlier [53].

### 2.7. ADMET Properties

Compounds with excellent pharmacological action are often rejected because of the difficulties encountered related to their metabolism and excretion as well as their high levels of absorption, distribution, and toxicity, collectively referred to as ADMET [54,55,56,57]. The selected derivatives were inspected based on drug-likeness (ADMET). The drug-likeness of a derivative were calculated by Lipinski’s rule of five (RO5) [58]. The calculation of physicochemical properties and ADMET properties were performed by ADMET lab 2.0 [59].

## 3. Results and Discussion

### 3.1. Chemistry

#### Synthesis of Phenylcarbamoylazinane-1,2,4-Triazole Amides Derivatives

The primary synthesis pathway for the formation of phenylcarbamoylazinane-1,2,4-triazole amides derivatives (**7a**–**o**) is illustrated in (Figure 2). The reactant: 4-(5-mercapto-4-phenyl-4H-1,2,4-triazol-3-yl)-N-phenylpiperidine-1-carboxamide (**4**) was treated for 30 min. with *N*-alkyl/aralky/aryl-2-bromoacetamides (**6a**–**o**) having different substituents. The mixture was then allowed to be refluxed for 4–5 h to get different products of phenylcarbamoylazinane-1,2,4-triazole amides derivatives (**7a**–**o**). The obtained product was further treated with the EtOH to obtain pure compounds as discussed in earlier work [27].

### 3.2. Density Functional Theory (DFTs)

#### 3.2.1. Molecular Geometry

The most stable optimized structural parameters, i.e., bond length, bond angle, and dihedral angles, were obtained implementing DFT/B3LYP/SVP calculations.

The gas phase optimized geometries further lead to solvent phase (ethanol) optimization and comparison. The DFT empowers to determine molecular properties such as optimized geometry and energy. The information of molecular geometry and molecular descriptors intended using quantum mechanical methods assist determination of molecular quantities distinguishing reactivity, shape, and binding properties of the selected molecules. The optimized structures of the most potent compounds (**7d** and **7f**) are shown in Figure 3 (below, and the rest of the optimized structures of compounds in gas phase and in solvent phase are given in Appendix A. Selected optimized geometrical parameters of triazole derivatives generated by using this method and they are presented in (Table 1) [31].

The dipole moment is a global assessment of the precision with which an electron density in a polar molecule is calculated. In this study, the results suggested that the dipole moment has an effect on a molecule’s interactions with other molecules as well as on electric fields [60]. Moreover, the dipole moment provides the basis for interpreting and quantifying intermolecular interactions. Another parameter, i.e., polarizability, is a critical characteristic in molecular electronics. To assess the precision of a quantum chemical approach, the electric characteristics of molecules provide the most direct relationships between the electronic structure of molecules and spectroscopically detectable values [61]. The optimized geometric characteristics of the targeted molecules in the gas phase and solvent phase (ethanol) are listed in Table 1.

#### 3.2.2. Frontier Molecular Orbital (FMOs)

The study of molecular orbitals and their energies is used to describe different kinds of reactions and to find the place in conjugated systems that is the most reactive. Molecular orbital energies, such as the highest occupied molecular orbital (HOMO) and the lowest unoccupied molecular orbital (LUMO), as well as their energy gaps, emphasize a point about the molecule’s biological and chemical activity. Having a small orbital gap suggests that a molecule is highly polarizable, which is often coupled with high chemical and kinetic reactivity and low kinetic stability. HOMO, or high-energy outer orbital containing electrons, works as an electron donor, and as a result, the ionization potential (I) is proportional to the energy of the high-energy outer orbital. However, LUMO may accept electrons, and its electron affinity (A) is proportional to the amount of energy it possesses [49].

In gas phase, **7k** had the smallest energy gap at 0.155 eV from all the selected triazole compounds. **7b** and **7d** had the higher energy gap in gas phase and shown the same value at 0.184 eV, among all the selected triazole compounds. **7e** and **7n** had slightly higher energy gap than **7k** at 0.158 eV and 0.160 eV, respectively.

In solvent phase (ethanol), **7j** had shown the smallest energy gap value at 0.178 eV. **7o** had only one point higher value of energy gap than **7j** at 0.179 eV. **7g** also had a comparable energy gap difference with value of 0.180 eV. Among all triazole compounds, **7d** had a higher energy gap difference at 0.193 eV. HOMO-LUMO structures of compounds **7d** and **7f** in gas phase are shown in Figure 3b and rest of the HOMO-LUMO structures of all compounds in gas phase and solvent phase are given in Appendix A.

#### 3.2.3. Global Chemical Reactivity Descriptors

By using HOMO-LUMO energy values, here the following parameters have been calculated by using their respective formulas:

Hardness: η = ½ (ELUMO-EHOMO); Chemical potential: µ = −χ; Softness: S = ½ η Electronegativity: χ = −½ (ELUMO + EHOMO); Electrophilicity index: ω = µ/2 η

Where A = −E _HOMO_ is the ionization potential and I = −E _LUMO_ is the electron affinity of the molecule. The estimated values such as E_HOMO_, E_LUMO_, ΔE_gap_, A, I, η, µ, S, χ and ω of triazole derivatives (**7a**–**o**) are given in Appendix A. As shown in (Appendix A), the compounds with the lowest energy gap were the compound **7k** and **7j** in gas phase and solvent phase (ethanol) (∆E_gap_ = 0.155 eV and 0.178 eV), respectively. This lower gap permitted it to be the softest molecules. The compounds with highest energy gap were the compounds **7b** and **7d** (∆E_gap_ = 0.184 eV), both showing the same value in gas phase. While in solvent phase (ethanol), **7d** (∆E_gap_ = 0.193) they exhibited the highest energy gap. The compound **7k** was the compound with the highest HOMO energy (E_HOMO_ = −0.207 eV), both in gas phase as well as in solvent phase (E_HOMO_ = −0.222 eV). These higher energy values make them the best electron donors. The compound with the lowest LUMO energy was **7o** (E_LUMO_ = −0.059 eV) in gas phase, while in solvent phase (ethanol) **7o** (E_LUMO_ = −0.046 eV) showed lowest LUMO energy, which predicted this derivative as the best electron acceptor. The two parameters, ionization potential (I) and electron affinity (A), are related to the one electron orbital energies of the HOMO and LUMO, respectively. The chemical reactivity of compounds differs with their structures. Chemical hardness and softness of compound **7k** (η = 0.077 eV, S = 6.47 eV) was best among all the compounds in gas phase. Thus, compound **7k** was assumed to be more reactive than all the compounds in gas phase. In solvent phase (ethanol), compound **7j** was found to be more reactive in comparison to all other compounds on the basis of values of hardness and softness (η = 0.089 eV, S = 5.61 eV) Compound **7o** in gas phase held a higher electronegativity value (χ = 0.143 eV) among all compounds, so it was the best electron acceptor. While in solvent phase (ethanol), **7f** and **7o** showed the same value of higher electronegativity (χ = 0.135 eV). Thus, both **7f** and **7o** were the best electron acceptors among all compounds in solvent phase (ethanol). Compound **7o** exhibited the higher value of electrophilicity index in gas phase as well as in solvent phase (ethanol) (ω_gas_ = 0.122 eV, ω_solvent_ = 0.103 eV) specifies this derivative as the stronger electrophiles among all. The compounds **7k** and **7j** both exhibited a lower frontier orbital gap; hence, they were found more polarizable, highly chemically reactive, had low kinetic stability, and were considered soft molecules.

#### 3.2.4. Molecular Electrostatic Potential (MEP)

The MEP is associated with the understanding of reactive sites of nucleophilic and electrophilic attacks, and ESP (electrostatic potential) correlates with a molecule’s partial charges and electronegativity properties. It is an essential tool for figuring out how a molecule is recognized as biologically active and how the electrophilic and nucleophilic attacks affect it. The surfaces and contours give a sense of how different geometries interact. The compounds’ electrostatic potential and electron density are depicted in Figure 3c. The figure showed that electron density is uniformly distributed throughout the molecules, and the ESP bar data showed that negative ESP is only localized in specific areas of a molecule as demonstrated by the experiment. Different colors in the ESP bar reflected the electrostatic potential values in Figure 3c rest of the electrostatic potential structures are shown in Appendix A. Highly negative electrostatic potential is illustrated by the red color, whereas the blue color is indicating highly positive potential and the green color is indicating the zero potential regions.

It is evident that the highly negative potential (red) interacted by withdrawing electrons is localized to nitrogen of the triazole ring and oxygen atom of carboxamide and phenylacetamide moiety. The findings are important to depict the nucleophilic and electrophilic attack on the molecule. As demonstrated in molecular docking, these areas were intimately engaged in interactions.

### 3.3. Molecular Docking

Density functional theory calculations and a detailed MEP analysis were used to find the best structure for the molecule. Electronegative parts of the molecule, such as the nitrogen of the triazole ring, the oxygen atom of the carboxamide, and the phenylacetamide moiety, were found, and their interactions in the active pocket were studied with molecular docking.

Crystal structures of both proteins reported the same substrate, with NAP attached to them. Docking of both targets with NAP assisted to compare the selected ligands (**7a**–**o**) with the co-crystal ligand. A conformation of the NAP with both targets is found in the Appendix A. Protein AKR1B1 has an active site with amino acids included indene system Tyr48, Lys77, and His110. The amino acids with benzene included Phe122 [62]. From the active pocket the most essential amino acids forming catalytic tetrad were Asp43, Lys77, His110, and Tyr48. Lys262, Arg268, and Asn272 are primarily involved in hydrogen bonding.

Structure-wise, the two proteins have a lot in common. They both have a Trp112 side-chain orientation that is not the same as the one in AKR1B10, which makes it possible for AKR1B1 inhibitors to keep their affinity for AKR1B10 by flipping Trp112 to make a AKR1B10 has an “AKR1B1-like” active site, whereas AKR1B10 inhibitors can take advantage of AKR1B10’s broad active site due to the native Trp112 side-chain orientation.

The binding energy values of the selected derivatives against both proteins, i.e., AKR1B1 and AKRB10, are given in Table 2.

#### 3.3.1. Structure Activity Relationship of Phenylcarbamoylpiperidine-1,2,4-Triazole Amide Derivatives

The structure activity relationship of these derivatives was studied on the basis of predicted inhibitory values obtained during docking and re-docking steps. An interesting behavior was observed in the case of both targeted protein results. The compounds that showed maximum binding interactions against the respective proteins were in correlation with the previously obtained anti-inflammatory activity results. Briefly, the derivatives **7d**, **7e**, **7f**, **7h**, **7j**, **7k**, and **7m** were reported as the best derivatives with micro-molar anti-inflammatory activity, and here in this study these derivatives exhibited maximum potential for the targeted proteins.

Against AKR1B1, among these derivatives, compound **7m** was found as the most potent derivative with the maximum binding interactions (binding score: −52.04 kJ/mol) and maximum (predicted) inhibitory constant value, i.e., 0.514 nM. When the predicted inhibitory constant values of this compound were compared to derivative **7d** (13.6 nM) and **7a** (53.8 nM), the substitutional effect was observed. The detailed structure activity compound **7d** having phenyl substitution showed better binding interactions (binding score: −49.4 kJ/mol) in comparison to **7a** (binding score: −42.36 kJ/mol)**,** indicating that the improvement in the activity might be due to the presence of the phenyl ring, which caused the inductive effect and enhanced the capacity of the compound to make strong interactions within the active pocket. In the case of **7m**, the presence of two methyl groups at 3 and 4 position of the phenyl ring activated the ring and caused the resonance effect, resulting in the stronger interaction as compared to **7a** and **7d**. The compound **7f** was found as the most potent inhibitor of AKR1B10 with a maximum binding score in comparison to the other derivatives, i.e., binding score: −52.84 kJ/mol and predicted inhibitory constant value i.e., 0.31 nM. This compound also showed good results against AKR1B1, i.e., binding score: −47.64 kJ/mol and predicted inhibitory value, i.e., 1.87 nM. The detailed structure-activity relationship of this compound suggested that the presence of methyl group was found favorable for the improved activity. This was compared with the activity of **7e**, **7j,** and **7k** derivatives and it was found that the enhanced activity of **7f** is due to the substitution at *meta* position.

Conclusively, the structure activity relationship of all the derivatives suggested that the substitution at *ortho* position alone is less favorable for the inhibition of either AKR1B1 or AKR1B10 as in case of **7e**, **7g**, and **7l** but the improved inhibition was found by those compounds where substitution was done at para alone (derivative **7h**) or in case of di-substitution, i.e., *ortho-para* (derivative **7j**), *ortho-meta* (derivative **7i**), or *meta-para* (derivative **7m**) position. Moreover, derivatives having *para* directing single substitution showed better results against AKR1B1 while compounds with *meta* alone (derivative **7f**) or di-substituted *meta-para* directing and *meta-meta* (derivative **7n**) directing substitution showed better results against AKR1B10. Figure 3 below illustrates the structure activity relationship of potent derivatives of AKR1B1 and AKR1B10 via docking studies.

#### 3.3.2. Detailed Molecular Docking Discussion of Protein AKR1B1

The 3D and 2D interactions of the most potent derivative **7d** are shown in Figure 4. The interactions were observed as strong hydrogen bonding, van der Waals interactions, π-interactions, which included π-π, π-cation, and π-anion interactions.

Significant hydrogen bonding was observed by nitrogen of ligand **7d** and amino acid residue of active pocket Lys262. Another hydrogen bonding was formed between oxygen and sulfur of N-benzyl-2(methylthio) acetamide of ligand **7d** and amino acid residue of active pocket Arg268. One of the predominant hydrogen bonding was found between the oxygen of N-phenylpiperidine-1-carboxamide of ligand **7d** and three amino acid residues (Gly213, Ser214 and Leu212). Electrostatic interaction (π-cation) was formed between toluene ring of ligand **7d** and Arg268.

In this paper, only potent compound (**7d**) is shown in Figure 4, all other compounds are discussed and shown with 3D and 2D conformations in Appendix A.

#### 3.3.3. Detailed Molecular Docking Discussion of Protein AKR1B10

The detailed 3D and 2D bonding and non-bonding interactions of the selected derivatives against AKR1B10 are shown in Figure 5. The interactions were observed as strong hydrogen bonding, van der Waals interactions, π-interactions, which includes π-π, π-cation, and π-anion interactions.

Sulfur of 2-mercapto-N-(*m*-tolyl) acetamide of ligand **7f** formed hydrogen bond complex with Lys22. Oxygen of N-phenylpiperidine-1-carboxamide of **7f** also formed hydrogen bonding interaction with Asn161 and Ser160. Another hydrogen bond was observed between piperidine ring of ligand **7f** and Tyr49. Electrostatic interaction (π-anion) was formed by one amino acid residue Asp217 with xylene of ligand **7f**. A favorable hydrophobic interaction (π-π T-shaped) was also observed by amino acid residue Trp21 with triazole ring of ligand **7f**.

In this paper, only potent compound **7f** is shown in Figure 5, all other compounds are discussed and shown with 3D and 2D conformations in Appendix A.

#### 3.3.4. SeeSAR Analysis

The SeeSAR analysis of the best compounds was executed, which anticipates the visual representation of binding interactions. The pharmacologically active compounds were presented as the blue-colored coronas, whereas the components having negative influence on binding interactions were shown as the red-colored coronas. The components with no role in contributing were shaded as colorless coronas. The size (small/large) of the corona indicated the structural component’s involvement. The SeeSAR representations of potent derivatives (Figure 6 and Figure 7), indicates that nearly the whole structure of **7d** was subsidizing favorably, but only phenyl urea group and two nitrogen of the base of triazoles ring were participating negatively (red coronas) because of the enormous energy of desolvation and the positive contribution denoted by green coronas. Same distribution of coronas presented in case of **7f**. The Hyde energies of the favorable coronas (green-colored) for derivative **7d** and **7f**, were −4.3 and −4.2 kJ/mol, respectively.

### 3.4. Molecular Dynamics Simulations

In order to investigate the dynamics and stability of proteins (AKR1B10 and AKR1B1) and the respective docked complexes, they were simulated in an aqueous environment. Proteins and their complexes were employed as starting points for MD simulations. For the analysis, the RMSD of the backbone of each system was determined with regard to their respective original configurations. The RMSD of both proteins and their complexes is shown in Figure 8a. The root mean square deviation of the AKR1B10 and AKR1B10−**7f** complexes varied initially but became steady after 60 ns. The average RMSD of the AKR1B10 and AKR1B10−**7f** complexes was determined to be 0.167 and 0.161 nm, respectively. The comparable result was observed for the complexes of AKR1B1 and AKR1B1−**7d**. The result demonstrates unequivocally the excellent stability of both proteins and their respective complexes in an aqueous solution. Additional analysis was performed using the RMSF formula, and the results are displayed in Figure 8b. The RMSF of the bulk of C atoms in the amino acid residues of the AKR1B10 and AKR1B10−**7f** complexes was less than 0.1 nm. Similarly, the RMSF of the majority of AKR1B1 and AKR1B1−**7d** residues was less than 0.1 nm. However, the terminal residues of both proteins exhibited greater oscillations, which could be a result of their suspended state. Additionally, the RMSF data demonstrated the stability of both proteins and their complexes in aquatic environments. Additionally, the RMSF of each atom in both ligands was computed (Figure 8c. Both ligands’ RMSFs fluctuated, indicating a dynamical alteration in their binding sites in respective proteins. The atoms of **7d** fluctuated more than those of **7f**.

The radius of gyration (R_g_), solvent accessible surface area (SASA), and energy of all systems were also analyzed by MD simulations. The R_g_ of the AKR1B10 and AKR1B10−**7f** complexes were found to be constant throughout the simulation and are therefore suggested to be stable. Finally, the stability of the compounds was established by computing their physicochemical properties, such as their potential and total energies, as shown in Figure 8c; (B). The total and potential energy of proteins and their complexes were found constant throughout the simulation, confirming the systems’ stability.

The hydrogen bond patterns of both ligands (**7d** and **7f**) were analyzed to determine their interaction with their respective proteins. Figure 9a illustrates the existence of hydrogen bonds between AKR1B10 and **7f** and AKR1B1 and **7d** at a concentration greater than 1%. As evidenced by the data, hydrogen bonds between ligands and their corresponding proteins existed indefinitely. Hydrogen bonds between **7f** and AKR1B10 were rather constant throughout the simulation time. However, the presence of hydrogen bonds between **7d** and AKR1B1 was greater until 65 ns, when it decreased marginally. The weakening of the hydrogen bond may be caused by the interaction of solvent molecules with the binding site [63]. Between AKR1B10 and **7f**, the average number of hydrogen bonds produced was 1.091. Similarly, an average of 1.602 hydrogen bonds was observed between AKR1B1 and **7d**. By computing the average of secondary structures of both ligands, the influence of their interaction on the secondary structural components of their respective proteins was studied. The Appendix A contains all of the information necessary to calculate secondary structures (Appendix A).

The PCA is a statistical approach that is used to characterize the large-scale motion of the biological macromolecules, during MD simulations. Moreover, it is used to reduce the dimension of a data collection without sacrificing critical information, which is represented as eigenvectors [64]. This analysis was used to determine both proteins’ flexibility in the absence as well as presence of respective ligands. Figure 9b illustrates the projection of proteins and complexes’ eigenvectors. The 2D projection showed that both complexes occupied nearly same conformational space as compared to their respective proteins alone, showing the presence of ligands did not alter the flexibility of the proteins. The free energy landscapes of AKR1B10 and AKR1B1 and their complexes was also made to examine the protein folding pattern, shown in Appendix A, and further discussion about free energy landscapes is also elaborated in Appendix A.

Moreover, the binding energies of **7f**-AKR1B10 and **7d**−AKR1B1 were calculated by using MM-PBSA analysis and further discussed in detail in Appendix A.

### 3.5. % Cell Viability Assay Using HeLa Cells

To strengthen the computational data, the most potent derivatives were tested using an in-vitro cell viability test, i.e., MTT assay. The human cervical cancer cells, i.e., HeLa cells, were treated at 100 µM concentrations for 24 and 48 h. This single-dose concentration was taken based on the predicted inhibitory values, obtained during molecular docking experiments. At a single dose concentration, a time-dependent linear response of cell death was found. Derivatives **7d** and **7f** demonstrated the maximum cell death, supporting the computational studies in which these molecules were reported as the strongest inhibitor of AkR1B1 and AKR1B10, respectively. Cisplatin was use as a positive control. The findings were computed by comparing them to the total activity control (i.e., untreated cells). Figure 10 is the depiction of % viability graph, created using GraphPad PRISM program.

### 3.6. ADMET Properties

The calculated physicochemical properties predicted that there was no contravention of Lipinski’s rule of five which was testimony of drug likeness of the compound. The details of each parameter of physicochemical and pharmacological properties are shared in Appendix A.

## 4. Conclusions

The results of molecular docking studies from both software revealed that our compounds showed better binding energy scores than the already reported anti-cancer compounds (vincristine, cisplatin) and the co-crystal ligand (NAP). However, the dispersion of the docking scores was seen, acquired from repeated docking runs. The docking program produced different docking results every time, under the same docking protocols. The large and bulky binding sites usually produce a great dispersion of the docking positions and scores, as there are more positions for the ligand conformations. It is not feasible to repeat calculations for the same ligands as it is a time-taking program. This is why we took the best poses from calculating the RMSD value. Additionally, for further validation and verification of our studies, control (vincristine) and co-crystal ligand (NAP) were docked along with other ligands. Thus, AutoDock was proved to be the best among the two software because it showed less dispersion as compared to MOE on repeated docking protocols. Then, DFT results showed the good chemical properties of the selected compounds after being optimized in gas phase as well as in solvent phase (ethanol). The ADMET properties showed that the selected compounds had drug-likeness properties. In addition to this, the results were supported by the cell viability assay.

Conclusively, from fifteen derivatives, only seven potent inhibitors (**7d**, **7e**, **7f**, **7h**, **7j**, **7k**, and **7m**) of both the targets were extracted after the deep analysis of docking scores and inhibition constant. In order to find the selective inhibitor of the selected targets, **7d** was identified as the best inhibitor of AKR1B1 and **7f** was proved as potent inhibitor of AKR1B10. Interestingly, the DFT results of the identified compounds were also in accordance with the other studies, suggesting these derivatives for the further investigation at molecular level and other pharmacological studies in future. Our study purposes to discover the potent aldo-keto reductase inhibitors that may help to synthesize effective molecule with drug-likeness for the treatment of colon cancer specifically occur from the abnormal expression of either protein AKR1B1 or AKR1B10. At last, the results of MD simulations confirmed the stability and dynamics of complex **7d** with AKR1B1 and complex **7f** with AKR1B10.

## Data Availability

Not Applicable.

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
