# Peer review of "Discovery of Phenylcarbamoylazinane-1,2,4-Triazole Amides Derivatives as the Potential Inhibitors of Aldo-Keto Reductases (AKR1B1 & AKRB10): Potential Lead Molecules for Treatment of Colon Cancer"

_molecules, 2022, doi:10.3390/molecules27133981_

Round 1

Reviewer 1 Report

My view remains the same as previous. Several similar computer-aided drugs discovery studies have been already published and there is no novelty in this study. Purely computational work usually requires experimental evidence to support the study's findings. Performing only cell viability (MTT) assay is not at all enough to evaluate the anticancer activity of selected compounds.

In my opinion, authors should perform in-vivo studies with some animal models to prove that their Phenylcarbamoylazinane-1,2,4-triazole amides derivatives are worth as inhibitors of aldo-keto reductases (AKR1B1 & AKRB10). This will validate the effectiveness of the in-silico method used as well as the author's recommendations. I can’t recommend the manuscript for publication in journal molecules.

Author Response

My view remains the same as previous. Several similar computer-aided drugs discovery studies have been already published and there is no novelty in this study. Purely computational work usually requires experimental evidence to support the study's findings. Performing only cell viability (MTT) assay is not at all enough to evaluate the anticancer activity of selected compounds.

Response: Yes the reviewer is right that computational work required experimental evidence. In our case based on available resources, we provided the anticancer effect of our compounds via MTT assay and for the further experiment we shall consider it in our upcoming publications. Hope the reviewer will understand our situation.

We did the extensive computational studies comprising density function theory studies which were used to correlate the electronic density stability and energy of compounds. Secondly, molecular docking studies for simulating and analyzing the molecular interactions of targeted protein and compounds. For the validation of stability and interpretation of molecular interactions of protein ligand complex via molecular dynamics simulation studies which is enough for supporting our results. We have also provided the in silico justification of our compounds as the best inhibitors of  NF-κB the results were found in correlation; 7d which was the best inhibitor of AKR1B1 was also found to be the best inhibitor of NF-κB (provided in supplementary file)

In my opinion, authors should perform in-vivo studies with some animal models to prove that their Phenylcarbamoylazinane-1,2,4-triazole amides derivatives are worth as inhibitors of aldo-keto reductases (AKR1B1 & AKRB10). This will validate the effectiveness of the in-silico method used as well as the author's recommendations. I can’t recommend the manuscript for publication in journal molecules.

Response: The reviewer gave a very fruitful suggestion and we agree that without in-vivo justification out results are partially validated. We have provided more in-silico justifications that our compounds have strong anticancer potential. Based on the previous comments, we have put purchase demand of our targeted Enzymes and it still in process. As In-vivo modeling will take time and at this stage authors of this study are thankful in advance to the reviewer to consider our new additions as justification of this point. We are working on more compounds and we will incorporate in-vivo studies in our upcoming publication along with the justification of current compounds.

Reviewer 2 Report

The Authors have investigated the influence of phenylcarbamoylazinane-1,2,4-triazole amides derivatives as the potential inhibitors of aldo-keto reductases. The manuscript deserves to publish in Molecules after a minor correction. I would like to suggest introducing changes before publishing in Molecules.

The authors should revise in the manuscript as the following points:

  1. Abstract: The abstract is too long. The abstract should state briefly the purpose of the research, the principle results and major conclusions. The abstract should be corrected.
  2. Why the Authors used B3LYP functional?
  3. Page 5 line 153: There is no information about the basis set used. Please complete the information.
  4. The article lacks confirmation of the structure of 7a-o compounds. Please complete this information in the supplementary materials.
  5. Throughout the article, energies should be expressed as kcal/mol or kj/mol.

Author Response

The Authors have investigated the influence of phenylcarbamoylazinane-1,2,4-triazole amides derivatives as the potential inhibitors of aldo-keto reductases. The manuscript deserves to publish in Molecules after a minor correction. I would like to suggest introducing changes before publishing in Molecules.

The authors should revise in the manuscript as the following points:

1. Abstract: The abstract is too long. The abstract should state briefly the purpose of the research, the principle results and major conclusions. The abstract should be corrected.

Response: Corrected as Suggested

2. Why the Authors used B3LYP functional?

Response: B3LYP is a functional that includes exact exchange and GGA corrections in addition to LDA electron-electron and electron-nuclei energy. The weights of the parts were fit to reproduce geometry of a test suite of small molecules. As such use of b3lyp for calculations with heavier atoms is questionable. B3LYP is generally faster than most Post Hartree-Fock techniques and usually yields comparable results. It is also fairly robust for a DFT method. On a more fundamental level, it is not as heavily parameterized as other hybrid functional, having only 3 where as some have up to 26. Becke's original paper is one of the most cited papers (I think it's number 8) of all time, so B3LYP is well established in the literature and people are less likely to complain about your choice versus a newer functional.

3. Page 5 line 153: There is no information about the basis set used. Please complete the information.

Response: Provided and highlighted as red text in the revised manuscript

4. The article lacks confirmation of the structure of 7a-o compounds. Please complete this information in the supplementary materials.

 Response: The reference of the Characterization data is provided in the manuscript and is also provided as separate “supplementary file only for review purpose” as suggested.  

5. Throughout the article, energies should be expressed as kcal/mol or kj/mol.

Response: Corrected as Suggested

Reviewer 3 Report

The manuscript entitled” Discovery of Phenylcarbamoylazinane-1,2,4-triazole Amides Derivatives  as the Potential Inhibitors of Aldo-Keto Reductases (AKR1B1 & AKRB10):
Potential Lead Molecules for Treatment of Colon Cancer” deals with mostly computational studies.But my opinion is that 1) also in vitro experiments for AKR1B1 and AKR1B10 aldehyde reductases  are needed  to demonstrate that anticancer activity is manifested through reduction of inflammation . 2) The compounds selected for computational studies are known for their action against 15-LOX. However, there is no good correlation and justification for their choice.

3) Furthermore , it is known that inhibitors of AKR1B1 & AKRB10 need an acidic group as a pharmacological element for binding at the active site of the enzymes and their potential inhibition action. This group is absent from these compounds and I doubt it they will be committed to the active center (as it is written in docking and dynamics) and especially because of their very large size. Finally, (4) I do not see in Manuscript the druglikeness and ADMET profile of compounds. Even authors mentioned something regarding supplementary material I did not see it.

Moreover, authors discussed detailed the Density functional theory (DFTs) and Molecular electrostatic potential saying that it is useful for docking, but they did not use these data. So what for they did all these? Authors should take care about English.

Minor.

Page 10.Lines 316, 317. Author twice repeat almost the same things. The compound 7k was with highest 316 HOMO energy (EHOMO=-0.207 eV) in gas phase. The highest HOMO energy in solvent 317
phase (ethanol) was of compound 7k (EHOMO=-0.222 eV)

Page 10.Line 325. Should be: was the best

Page 11.Line 348. Better to say: correlates

Pgae 12.Line 386. Should be: in correlation

Page 16. Lines 466,470, 471. Compounds should be in bold

In references 32, 38, 39 and 42 only the first pages is mentione.

Author Response

The manuscript entitled” Discovery of Phenylcarbamoylazinane-1,2,4-triazole Amides Derivatives  as the Potential Inhibitors of Aldo-Keto Reductases (AKR1B1 & AKRB10):
Potential Lead Molecules for Treatment of Colon Cancer” deals with mostly computational studies.But my opinion is that

1) also in vitro experiments for AKR1B1 and AKR1B10 aldehyde reductases  are needed  to demonstrate that anticancer activity is manifested through reduction of inflammation

Response: Yes the reviewer gave the wonderful suggestion and this we have already planned for our upcoming publication and both enzymes are already in purchase demand. All the authors are thankful to the reviewer in advance for considering this study based on computational data. Another justification of role of these derivatives in NF-κB has been incorporated in the revised supplementary file.

2) The compounds selected for computational studies are known for their action against 15-LOX. However, there is no good correlation and justification for their choice.

Response: The rationale behind the selection of these compounds is based on their anti-inflammatory effect via 15-LOX enzyme inhibition assay and molecular docking studies. As our targeted enzyme AKR1B1 involves in inflammation in cancer via NF-κB pathway therefore we selected these derivatives in order to find the best lead molecule (as an antagonist) for AKR1B1.

3) Furthermore , it is known that inhibitors of AKR1B1 & AKRB10 need an acidic group as a pharmacological element for binding at the active site of the enzymes and their potential inhibition action. This group is absent from these compounds and I doubt it they will be committed to the active center (as it is written in docking and dynamics) and especially because of their very large size.

Response: Yes the reviewer is right, the acidic part is inevitable for activity of inhibitors of AKR family. But it is observed in our study that 1,2,4-triazole ring (NH-protons in N-unsubstituted-1,2,4-triazoles are acidic in nature [https://doi.org/10.1016/B978-0-12-409547-2.14854-1]) and carboxamide moiety is highly involved in significant hydrophilic and important hydrophobic interactions with the active site amino acid residues, which lend enough testimony to this comment

Finally, (4) I do not see in Manuscript the druglikeness and ADMET profile of compounds. Even authors mentioned something regarding supplementary material I did not see it.

Response: Due to the limitation of words, we placed ADMET result in supplementary file, results are reported in result and discussion part under sub-heading 3.6.

Moreover, authors discussed detailed the Density functional theory (DFTs) and Molecular electrostatic potential saying that it is useful for docking, but they did not use these data. So what for they did all these? Authors should take care about English.

Response: The corrections have been incorporated in the respective parts of revised manuscript, as suggested

Minor.

Page 10.Lines 316, 317. Author twice repeat almost the same things. The compound 7k was with highest 316 HOMO energy (EHOMO=-0.207 eV) in gas phase. The highest HOMO energy in solvent 317
phase (ethanol) was of compound 7k (EHOMO=-0.222 eV)

Response: Corrected as suggested

Page 10.Line 325. Should be: was the best

Response: Corrected as suggested

Page 11.Line 348. Better to say: correlates

Response: Corrected as suggested

Pgae 12.Line 386. Should be: in correlation

Response: Corrected as suggested

Page 16. Lines 466,470, 471. Compounds should be in bold

Response: Corrected as suggested

In references 32, 38, 39 and 42 only the first pages is mentione.

Response: Corrected as suggested

Round 2

Reviewer 1 Report

The manuscript submitted by Syeda Abida Ejaz and Jameshed Iqbal deals with the discovery of phenylcarbamoylazinane-1,2,4-triazole amides derivatives as the aldo-Keto reductases inhibitors.

This referee agrees with the main purpose of this work, as extensive computational study and their molecular models might help in the design of efficient anticancer drugs.

It is not my intention to slow down the publication. As other referees agree with the manuscript content and by considering the available resources authors have to perform animal model studies, I would like to recommend the manuscript for publication as it is.

I am very happy with the author's polite reply for accepting the fact that pure computational work required experimental evidence like animal model studies. The authors are very good at computational work and please consider using animal models in future publications. I wish all the best for your publication in ‘Molecules’.

Author Response

Comment: This referee agrees with the main purpose of this work, as extensive computational study and their molecular models might help in the design of efficient anticancer drugs. It is not my intention to slow down the publication. As other referees agree with the manuscript content and by considering the available resources authors have to perform animal model studies, I would like to recommend the manuscript for publication as it is.

Response:  The authors are very thankful to the reviewer for understanding the situation.

I am very happy with the author's polite reply for accepting the fact that pure computational work required experimental evidence like animal model studies. The authors are very good at computational work and please consider using animal models in future publications. I wish all the best for your publication in ‘Molecules’.

Response:  We appreciate your kind words

Reviewer 3 Report

The manuscript entitled” Discovery of Phenylcarbamoylazinane-1,2,4-triazole Amides Derivatives  as the Potential Inhibitors of Aldo-Keto Reductases (AKR1B1 & AKRB10):
Potential Lead Molecules for Treatment of Colon Cancer” deals with mostly computational studies. Authors did not answer and followed my suggestion. They only respond to grammar and language mistakes as well as my comments regarding NF-kB. I would like to say that docking is only prediction and need justification by experimental evaluation. But taking into account the topic of this special issue I will accept author’s answers. I should mentioned that I found plagiarism and mentioned this in my comments in the first round of revision process authors did not take this into account. Manuscript still have plagiarism.

Author Response

Comment:. Authors did not answer and followed my suggestion. They only respond to grammar and language mistakes as well as my comments regarding NF-kB. I would like to say that docking is only prediction and need justification by experimental evaluation. But taking into account the topic of this special issue I will accept author’s answers. I should mentioned that I found plagiarism and mentioned this in my comments in the first round of revision process authors did not take this into account. Manuscript still have plagiarism.

Response: The grammatical, as well as typographic mistakes, have been omitted and revised throughout the manuscript. Moreover, the plagiarism has been removed, as suggested.

This manuscript is a resubmission of an earlier submission. The following is a list of the peer review reports and author responses from that submission.

Round 1

Reviewer 1 Report

Authors suggested phenylcarbamoylazinane-1, 2,4-triazole amides derivatives (7a-o) as the inhibitors of aldo-keto reductases (AKR1B1, AKR1B10) via computational analysis. Several such computer-aided drug discover studies have been already published.

Purely computational work usually requires experimental evidence to support the study's findings. Authors must perform in-vitro and in-vivo studies to prove that their Phenylcarbamoylazinane-1,2,4-triazole amides derivatives are worth as inhibitors of aldo-keto reductases (AKR1B1 & AKRB10). This will validate the effectiveness of the in-silico method used as well as the author's recommendations. This is necessary.

Reviewer 2 Report

The Authors have performed the study which was focused on the identification of phenylcarbamoylazinane-1, 2,4-triazole amides derivatives as the inhibitors of aldo-keto reductases via detailed computational analysis. The manuscript deserves to publish in Molecules after a major correction. I would like to suggest introducing changes before publishing in Molecules.

The authors should revise in the manuscript as the following points:

  1. Abstract: The abstract should state briefly the purpose of the research, the principle results and major conclusions. The abstract should be corrected.
  2. Page 5, line 135: it should be: “The 1H and 13C NMR spectra were operated..”
  3. Page 5, point 2.3: the tool used in the DFT should be described in detail.
  4. Designations throughout the article should be standardized, e.g. h and hrs.
  5. Scheme 2: this scheme is hardly legible. Please make the table with the R substituents and yields.
  6. Why did the authors carry out calculations in a methanol environment, since they used ethanol for the reaction?
  7. Point 3.2.3 - in the application the authors forgot to send supplementary materials, this part cannot be checked unfortunately.
  8. The energy barriers in the article were given in various units in kJ / mol and kcal / mol. Please standardize the units.
  9. Correct figure 5 as it is not legible.